# Yield Performance and Physiological Response of a Maize Early Hybrid Grown in Tunnel and Open Air under Different Water Regimes

**Lucia Ottaiano** [1], **Ida Di Mola** [1,*], **Chiara Cirillo** [1,*], **Eugenio Cozzolino** [2] and **Mauro Mori** [1]

1   Department of Agricultural Sciences, University of Naples Federico II, 80055 Portici, Italy;
    lucia.ottaiano@unina.it (L.O.); mori@unina.it (M.M.)
2   Council for Agricultural Research and Economics (CREA)—Research Center for Cereal and Industrial Crops,
    81100 Caserta, Italy; eugenio.cozzolino@crea.gov.it
*   Correspondence: ida.dimola@unina.it (I.D.M.); chiara.cirillo@unina.it (C.C.)

**Abstract:** Climate change is one of the most important and studied phenomena of our age and it can have a deep impact on agriculture. Mediterranean countries are and will continue to be strongly affected by changing environmental factors, including lack of precipitation and prolonged heatwaves. The current study aimed to assess the adaptability of an early maize hybrid grown in two temperature conditions and subjected to different irrigation water regimes. The experimental design was a randomized complete-block design with two different temperature conditions: (i) ordinary temperature in open field (OF) and (ii) high temperature (about 3 °C higher than the current condition) under a poly-ethylene tunnel (PE). In both environments, five irrigation level treatments were applied: 100% (DI100), 75% (DI75), 50% (DI50), 25% (DI25), and 0% restoration of water lost by evapotranspiration (DI0). The responses of maize plants were assessed in terms of yield, nitrogen content determination, nitrogen use efficiency, leaf gas exchanges, and leaf water potential measurements. In both conditions, yield and its components linearly decreased as the irrigation water amount reduced, and even the DI0 plants did not produce. Notably, the PE-DI100 treatment had a significantly higher yield than the corresponding treatment in the open air (9.9 vs. 8.5 t ha$^{-1}$), due mainly to the increased number of ears per square meter (13 vs. 11 m$^2$, respectively). Though, as far as it concerns physiological parameters, a significant effect of environmental conditions was found, with values significantly lower under the protected environment, compared to the plants in the open field. Considering our results, it can be assumed that correct management of amount and time intervals of irrigation could adapt the maize to future climate change.

**Keywords:** heat stress; water deficit; maize; yield; water use efficiency; photosynthetic activity

## 1. Introduction

Climate change is one of the most important and studied phenomena of our age. It involves deep changes, especially in the temperature, rain, and rise in atmospheric carbon dioxide concentration ($CO_2$). The Earth's increasing temperature is one of the most obvious results of climate change, with estimated global warming of 1.5 °C with respect to pre-industrial levels [1]. Ganachaud et al. [2] reported that the temperature will be increased up to 0.7–0.8 °C by 2035 and 2.3–3.0 °C by 2100. In the Mediterranean, it is assumed that the warming will exceed the global rate by 20%, and there will be a reduction of precipitation over most of its areas [3].

These rising temperatures are expected to bring about an increase in the frequency of heatwaves and precipitation variable patterns in most of the world areas. Temperature is one of the most important factors that influence seed viability and seedling growth, germination capacity and rate [4], as well as processes that affect plant biomass production and fruit and grain yield [5]. High average "seasonal" temperature can increase the risk of

drought, limit photosynthesis rates, and reduce light interception by accelerating phenological development. In Italy, Tubiello et al. [6] described that warmer temperatures accelerated plant phenology and reduced dry matter accumulation, with an impact on yields in the order of 20%. They also found that the maize growing cycle was shortened by 16 days with respect to the 114 days of the ordinary cycle length, and also evapotranspiration was diminished by 70 mm.

Water availability is another crucial factor affecting crop production. Indeed, many studies reported high sensitivity to drought and other environmental stresses in maize [7,8]. Irrigation influences maize grain yield, and to obtain the highest yield, complete restitution of crop evapotranspiration is needed. As reported by English and Raja [9], the crucial aim of appropriate irrigation water application is to increase water productivity. The irrigation scheduling allows for maximizing crop yield and using efficiently scarce water resources. Some studies report that corn appears to be relatively tolerant to water deficits during the vegetative and ripening periods, and that the greatest decrease in grain yield is caused by a moisture deficit along the soil profile during the flowering period [8,10,11]. Short-term effects of water deficits are described as delayed leaf tip emergence and leaf area reduction. Long-term water deficits may cause a reduced final size of the leaves and internodes, and yield losses of 15–25% [12]. Some authors reported that water deficits reduce crop growth, canopy development, and morphological characteristics of the corn plant, and reduce dry matter accumulation [13,14].

Some research reported that mild to moderate drought stress causes problems to stomatal characteristics, resulting in biomass loss [15]. Such losses depend on the time of the plant cycle in which water stress occurred [16], which reduces the photosynthetic rate and $CO_2$ fixation [17].

When drought conditions persist, reactive oxygen species (ROS), such as hydroxyl radicals (OH), superoxide ($O_2^-$), hydrogen peroxide ($H_2O_2$), and singlet oxygen, are generated [18], which induces oxidative damage due to increased production of ROS in plants [19–21].

Furthermore, it was shown that the water stress and high-concentration $CO_2$ decrease stomatal conductance (Gs) and cause a higher photosynthetic rate (An). In the literature, it is reported that gas exchange in plants takes place through stomata, while opening and closing of stomata depends on diverse environmental factors, including $CO_2$ concentration and water availability in the soil [22].

Agriculture, in the broadest sense, will face new problems as a consequence of rising temperatures. Mediterranean countries are and will continue to be strongly affected by changing environmental factors, including lack of precipitation and prolonged heatwaves. Since the availability of maize hybrids able to counteract multiple stresses, such as increasing air temperatures and drought under current changing environmental conditions, and related mechanisms involved, is under investigation, the responses of maize genotypes already in cultivation need to be deeply investigated.

The aim of the current experiment was to evaluate the adaptability of an early maize hybrid grown in two temperature conditions (ordinary and high) and subjected to different water regimes.

## 2. Materials and Methods

### 2.1. Experimental Site and Design, Crop Management, and Crop and Yield Measurements

The experiment was set up in 2017 at the experimental site of the Department of Agricultural Science, in Portici (Naples, Italy; latitude 40°49' N; longitude 14°20' E).

A randomized complete-block design was used, and five replicates per treatment were made. The crop was subjected to two different temperature conditions: (i) ordinary temperature in open field (OF) and (ii) high temperature under a polyethylene tunnel (PE). In both environments, five water treatments were applied: 100% (DI100), 75% (DI75), 50% (DI50), 25% (DI25), and 0% restoration of water lost by evapotranspiration (DI0). The water loss was calculated by the Hargreaves method, on a 5–7-day basis, and deducing

rainfall, the crop coefficient (Kc) changed as a function of the phenological phase. The water quantity applied during the whole cycle is reported in Table 1: the open field DI0 treatment had only the rainfall water (about 45 m$^3$ ha$^{-1}$ during the crop cycle) and the same quantity was also used in the corresponding tunnel treatment.

The water use efficiency (WUE, expressed as kg m$^{-3}$) was calculated according to the following formula:

$$WUE = \frac{Yield}{Total\ water\ (irrigation + rainfall)} \tag{1}$$

**Table 1.** Water use of the irrigation treatments (DI100 = 100%, DI75 = 75%, DI50 = 50%, DI25 = 25%, and DI0 = 0% restoration of water lost by evapotranspiration) during the whole cycle under the two environments (PE = high temperatures and OF = ordinary temperatures).

| Irrigation | Water Use (m$^3$ ha$^{-1}$) | |
|---|---|---|
| | PE | OF |
| DI100 | 3964.4 | 3314.9 |
| DI75 | 2978.5 | 2497.5 |
| DI50 | 1997.3 | 1680.1 |
| DI25 | 1011.4 | 862.7 |
| DI0 | 45.3 | 45.3 |

The used crop was an early hybrid maize (*Zea mays L.*, class FAO 200), which was sown on 3 April. The crop was cultivated in plastic pots of 0.33 cm$^2$ with a plant density of 3 plants per pot. Soil was loamy sand (USDA), with a good fertility (Table 2).

Nitrogen was provided at the dose of 160 kg ha$^{-1}$ as ammonium nitrate (26% *n*) and it was applied two times: 1/3 at the sowing and 3/4 at the 6-leaf stage. Phosphorus and potassium were not applied because the soil contents were high.

The harvest was performed two times in the first week of July.

**Table 2.** Physical and chemical properties of the test soil.

| Soil Properties | Units | Mean Values |
|---|---|---|
| Coarse sand | % | 38.6 |
| Fine sand | % | 40.5 |
| Silt | | 14.3 |
| Clay | | 6.6 |
| N—total (Kjeldahl method) | % | 0.16 |
| P$_2$O$_5$ (Olsen method) | ppm | 312.8 |
| K$_2$O (Tetraphenylborate method) | ppm | 620.7 |
| Organic matter (Bichromate method) | % | 3.57 |
| NO$_3$-N | ppm | 32.73 |
| NH$_4$-N | ppm | 6.05 |
| pH | | 7.14 |
| Electrical conductivity | dS m$^{-1}$ | 0.32 |

### 2.2. Crop Growth and Yield Measurements, Nitrogen Content Determination, and Nitrogen Use Efficiency

During the cycle, the crop growth was monitored by four biometric samplings in order to measure the following parameters: height of stalks, leaves and ears number, and fresh weight of each part. Then, a sample per each treatment and replicate was oven-dried at 60 °C until reaching constant weight, in order to determine the dry matter.

Leaf area index was measured with an electronic leaf area meter (Li-Cor3000, Li-Cor, Lincoln, NE, USA).

At the harvest, the biomass was cut, and its components (stalks, leaves, and ears) were separately weighed and then oven-dried at 60 °C until reaching constant weight. The yield

was expressed as ton per hectare. In addition, basal diameter, length, and the percentage of fertile part of ears (ear length with filled grains) were measured.

On a representative sub-sample of kernels for each replicate, nitrate and total nitrogen content were determined by chemical analysis based on the colorimetric and Kjeldhal methods [23], respectively. The protein content of kernels was determined by multiplying nitrogen content by the factor of 6.25.

Nitrogen use efficiency was calculated dividing fresh yield by N applied [24].

## 2.3. Leaf Gas Exchanges and Leaf Water Potential Measurements

At 71 DAS, measurements of leaf gas exchange were conducted within 2 h across solar noon (i.e., between 11:00 and 13:00 h) on the youngest fully expanded leaves, using nine replicates per treatment, as described by Cirillo et al. [25]. Briefly, the net $CO_2$ assimilation rate (Pn), stomatal conductance (gs), and transpiration rate (E) were determined with a portable gas-exchange analyzer (LCA 4; ADC BioScientific Ltd., Hoddesdon, UK), equipped with a broad-leaf PLC (cuvette window area, 6.25 cm$^2$). Photosynthetically active radiation, relative humidity, and carbon dioxide concentrations were set at ambient value and the flow rate of air was 400 mL s$^{-1}$. Leaf intrinsic water use efficiency (WUEi) was calculated as the ratio of net $CO_2$ assimilation rate (Pn) to stomatal conductance to water vapor (gs).

Leaf water potential ($\Psi_1$) was measured at the same date of the gas exchange measurements (71 DAS) by selecting one well-lit leaf per plant on three plants per replication per treatment, as reported by Carillo et al. [26]. Leaf water potential was measured using the pressure chamber (3005-series portable plant water status console, Soil Moisture Equipment Corp., Santa Barbara, CA, USA) technique [27], using the precautions proposed by Turner and Long [28]. The leaf water potential measurements were performed in the morning, at midday, and in the afternoon (i.e., 9:00 h, 12:00 h, 17:00 h).

## 2.4. Statistical Analysis

All data were subjected to the analysis of variance (one-way ANOVA), using a general linear model by the SPSS software package (SPSS version 22, Chicago, IL, USA). Means were separated according to the Duncan test at $p \leq 0.05$.

## 3. Results and Discussion

### 3.1. Climate Characteristics of Experimental Site

During the whole growing period, the minimum air external temperature was similar to the tunnel one, ranging between 4 °C in April and 18 °C in June (Figure 1). The maximum temperature trend was different; indeed, under the tunnel, it was on average 3.3 °C higher than the external temperature (Figure 1), overcoming, from the end of May and until the harvests, 33 °C, limiting the temperature for optimal growth and flowering. Instead, in the open air, only from the middle of June was this temperature overcome.

In particular, during the first growing phase (from sowing to the first ten days of May), the average air temperature was 4.1 °C higher under tunnels compared to the field; instead, in the other phases of the cycle, this difference was reduced to about 2.5 °C. This suggests that in cold periods, the tunnel is able to keep the temperature more constant, reducing the negative effect of low temperatures. Indeed, Jayalath et al. [29], in their research carried out in Georgia during spring, also reported an average temperature increase ranging from 0.5 to 1.6 °C in protected conditions (tunnel) compared to the open air; in addition, they detected that on the coldest mornings of the experiment, the tunnel maintained temperatures 3.0 to 5.0 °C higher than the field, highlighting its ability to mitigate cold temperatures.

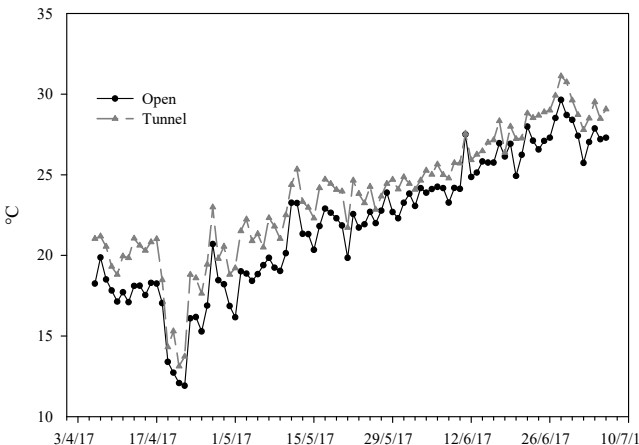

**Figure 1.** Average temperature during the test period in the two environments: open (OF) and tunnel (PE).

### 3.2. Growth Parameters as Affected by Temperature and Water Stress

The effect of the five water treatments on total dry matter accumulation is reported in Figure 2A,B.

In both environments (OF and PE), as expected, the trend was increasing, reaching the maximum value in mid-June at the flowering, when the first statistical differences between the water treatments were recorded. Generally, the dry matter accumulation decreased at the increase of water stress; in OF, the percentage decrease over DI100 was 20%, and 56%, 71%, and 95%, for DI75, DI50, DI25, and DI0, respectively (Figure 2B); whereas, in PE, the percentage decreases were 29%, 50%, 72%, and 82%, for DI75, DI50, DI25, and DI0, respectively (Figure 2A). Interestingly, under high temperatures (PE), the decrease in dry matter accumulation in DI75 plants was higher than in the corresponding treatment under OF conditions. It is possible that the higher temperatures of PE exacerbated the reduction in dry matter accumulation due to water stress. On the other hand, in other studies, it was also observed that high air temperatures had a negative effect on the maize growth, due to the increased evapotranspiration and water stress, with a consequent reduction of photosynthetic activity and therefore a decline in plant growth [30–32].

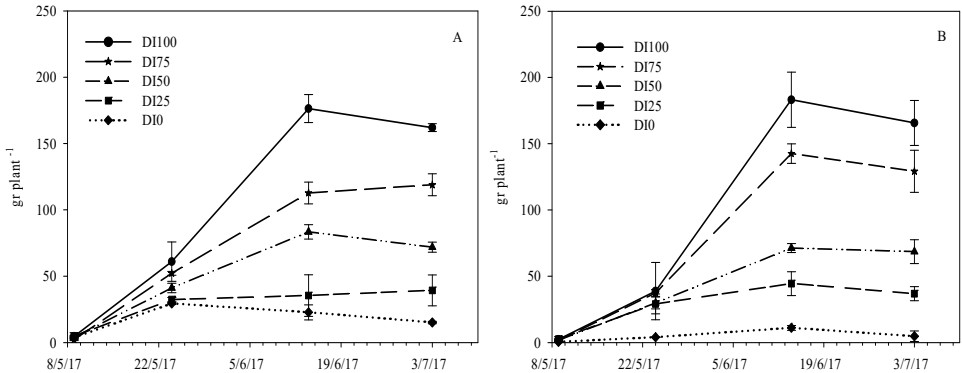

**Figure 2.** Accumulation of total dry matter during the crop cycle as affected by five irrigation levels (DI100 = 100%, DI75 = 75%, DI50 = 50%, DI25 = 25%, and DI0 = 0% restoration of water lost by evapotranspiration) in the two environments (PE = high temperatures (**A**) and OF = ordinary temperatures (**B**)).

Plant height was also affected by water treatments in both environmental conditions (Figure 3). Obviously, the height always showed an increasing trend, and it reached the highest value at the flowering stage (second decade of June). In PE, the plants of the DI100 and DI75 treatments reached 150 and 133 cm respectively, and they were different between

them (Figure 3A). In OF, the height of DI100 and D75 plants was lower (135 and 117 cm, respectively), and they were also different between them (Figure 3B). The differences between the two environments are most likely due to the fact that the high temperatures cause the elongation of plants.

For all other treatments, the height values were similar in the two environments (D50 = 88 cm, D25 = 58 cm, and D0 = 36 cm on average), however, per each environment, they were different between them starting from flowering (Figure 3A,B).

These differences in height between treatments could be due to differences in cell expansion and turgor, as reported by other authors who found that water stress can reduce cell expansion through reduced turgor pressure and a reduction of cell division [33,34].

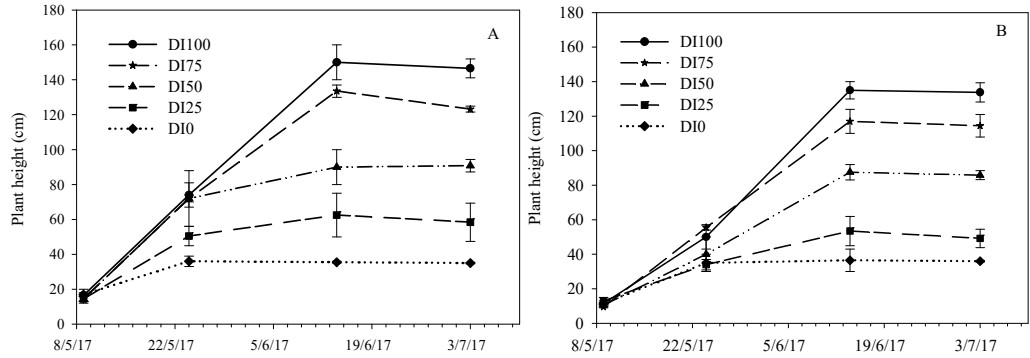

**Figure 3.** Plant height during the crop cycle as affected by five irrigation levels (DI100 = 100%, DI75 = 75%, DI50 = 50%, DI25 = 25%, and DI0 = 0% restoration of water lost by evapotranspiration) in the two environments (PE = high temperatures (**A**) and OF = ordinary temperatures (**B**)).

The effects of the interaction between environmental conditions and deficit irrigation application on green leaves number, leaf area index (LAI), and average leaf area at flowering are reported in Table 3. The highest values of leaves number were recorded in OF, with 12 vs. 10.6 of PE, and all treatments of the two environments were different between them, except DI100 and DI0. Additionally, for LAI, higher values were observed under the open field condition compared to high-temperature conditions (3.5 vs. 2.3 $m^2\ m^{-2}$) but with a different trend. The LAI values of DI100 and DI75 plants were not statistically different in the two environments, conversely to the other irrigation treatments (DI50 and DI25), which were statistically different. Regarding the leaf area, a different trend was recorded; indeed, the treatment DI100-PE was statistically higher than all other treatments, a result of a statistically higher average leaf area, which allows compensating for the lower leaves number. Additionally, DI75 showed a higher value of LA with respect to OF, but without significant differences between the two environments; instead, the trend was inverted for the three most stressed treatments (DI50, DI25, and DI0), and for the last two, the same trend was recorded for the average leaf area. This confirms that one of the adaptive responses of plants is the reduction of the LA, which is due not only to the reduction of leaves number but also to the reduction of the leaf size.

Our results are consistent with the findings of previous studies, where some authors analyzed the effects of different water stresses in maize and observed significant delays in leaf tip emergence and reductions in leaf expansion [34–36].

**Table 3.** Interaction between environmental conditions (PE = high temperatures and OF = ordinary temperatures) and deficit irrigation application (DI100 = 100%, DI75 = 75%, DI50 = 50%, DI25 = 25%, and DI0 = 0% restoration of water lost by evapotranspiration) on green leaf number, LAI (leaf area index), LA (leaf area), and ALA (average leaf area) at flowering.

| Irrigation | Leaves (n pt$^{-1}$) | | LAI (m$^2$ m$^{-2}$) | | LA (cm$^2$ pt$^{-1}$) | | ALA (cm$^{-2}$ leaf$^{-1}$) | |
|---|---|---|---|---|---|---|---|---|
| | PE | OF | PE | OF | PE | OF | PE | OF |
| DI100 | 10.5 b | 11.0 b | 4.0 ab | 3.9 ab | 281.7 a | 263.2 b | 26.8 a | 23.9 b |
| DI75 | 11.0 b | 12.0 a | 3.9 ab | 4.1 a | 260.2 bc | 253.0 c | 23.7 b | 21.1 c |
| DI50 | 10.5 d | 12.0 a | 2.8 d | 3.8 b | 194.5 e | 235.5 d | 18.5 de | 19.6 cd |
| DI25 | 10.5 d | 12.0 a | 1.8 e | 3.1 c | 127.0 g | 191.6 e | 12.1 f | 16.0 e |
| DI0 | 7.5 c | 7.5 c | 1.3 f | 1.6 e | 121.3 g | 160.4 f | 16.2 e | 21.4 bc |
| **Significance** | | | | | | | | |
| Irrigation (DI) | * | | ** | | ** | | * | |
| Environment (E) | * | | * | | * | | * | |
| DI × E | ** | | * | | ** | | * | |

NS, * and **, refer to not significant or significant at $p \leq 0.05$, and 0.01, respectively. Different letters within each column indicate significant differences according to Duncan's multiple range test.

At the harvest, percentage incidence of the different parts of plants in the total dry matter was determined, and the data are reported in Tables 4 and 5, for PE and OF, respectively. In both environments, the plant senescence increased when the water stress increased, as highlighted by the higher incidence of yellow leaves in the most stressed treatments. Indeed, for the DI0-PE, the presence of green leaves was not recorded, and the percentage of yellow leaves reached 83.1% (Table 4). The same treatment, in the open field, showed about 14% of green leaves and 53% of yellow leaves in total dry matter (Table 5). Therefore, as conceivable, it seems that the higher temperatures of the PE condition further stressed the senescence process. For all treatments, the percentage incidence of the ear was higher in the tunnel (44.6% vs. 39.5%, respectively) (Tables 4 and 5). As regards the irrigation level, the treatment DI50 showed a halved percentage incidence of ears compared to DI100 in both environments (Tables 4 and 5).

**Table 4.** Percentage incidence of different parts of plants in total dry matter for the five irrigation levels (DI100 = 100%, DI75 = 75%, DI50 = 50%, DI25 = 25%, and DI0 = 0% restoration of water lost by evapotranspiration) in the PE = high temperatures.

| Irrigation | Stem | Green Leaves | Yellow Leaves | Ears |
|---|---|---|---|---|
| | | **% Dry Matter** | | |
| DI100 | 23.70 b | 10.45 a | 2.65 c | 63.20 a |
| DI75 | 27.00 b | 11.63 a | 3.31 c | 58.06 a |
| DI50 | 45.60 a | 8.28 a | 13.56 b | 32.57 b |
| DI25 | 45.20 a | 11.70 a | 18.60 b | 24.51 c |
| DI0 | 16.86 c | 0.00 b | 83.14 a | 0.00 d |
| **Significance** | | | | |
| Irrigation (DI) | ** | ** | ** | ** |
| Environment (E) | NS | NS | NS | NS |
| DI × E | NS | NS | NS | NS |

NS, ** refer to not significant or significant at $p \leq 0.01$, respectively. Different letters within each column indicate significant differences according to Duncan's multiple range.

**Table 5.** Percentage incidence of different parts of plants in total dry matter for the five irrigation levels (DI100 = 100%, DI75 = 75%, DI50 = 50%, DI25 = 25%, and DI0 = 0% restoration of water lost by evapotranspiration) in the OF = ordinary temperatures.

| Irrigation | Stem | Green Leaves | Yellow Leaves | Ears |
|---|---|---|---|---|
| | | **% Dry Matter** | | |
| DI100 | 26.35 c | 9.72 b | 5.75 c | 58.18 a |
| DI75 | 29.16 bc | 7.05 b | 8.92 c | 54.88 a |
| DI50 | 43.34 a | 10.19 b | 16.48 b | 30.00 b |
| DI25 | 47.73 a | 16.80 a | 20.46 b | 15.01 c |
| DI0 | 33.10 b | 13.88 a | 53.02 a | 0.00 d |
| **Significance** | | | | |
| Irrigation (DI) | ** | ** | ** | ** |
| Environment (E) | NS | NS | NS | NS |
| DI × E | NS | NS | NS | NS |

NS, ** refer to not significant or significant at $p \leq 0.05$ respectively. Different letters within each column indicate significant differences according to Duncan's multiple range test.

### 3.3. Maize Performance: Yield and Its Components, and Water Use Efficiency (WUE)

The results regarding yield and its components (plant height, number, basal diameter, total length, and fertile part of the ears) of maize are reported in Tables 6 and 7. The interaction between the temperature conditions and water irrigation level was found for all parameters. In both conditions (OF and PE), all parameters linearly decreased as the irrigation water amount reduced, and even the DI0 plants did not produce, and they only reached a height of 36 cm. Notably, the DI100 treatment under the tunnel (high temperature) had a significantly higher yield than the corresponding treatment in the open air (9.9 vs. 8.5 t ha$^{-1}$), due mainly to the increased number of ears per square meter (13 vs. 11 m$^2$, respectively) (Table 6). Indeed, the percentage of ear length with filled grains (fertile part (F.P.)) for all deficit irrigation treatments was slightly higher in plants grown under open field conditions (Table 7), probably because the higher temperatures reached under the tunnel determined flowering and grain ripening disorders. This result is consistent with that found by Shiferaw et al. [37], who reported that temperatures beyond 40 °C, mainly during flowering and grain filling, have a severe impact on plant grain productivity. In addition, also in other crops, such as snap bean, similar effects of high temperatures during flowering and fruit ripening on plant growth and yield were recorded [38]. However, in both temperature conditions, the percentage of fertile ear decreased with increasing water stress; indeed, the less stressed treatments (DI100 and DI75) exceeded the 73% F.P. with an average value of 75.7%, and the DI25 plants in both environments had less than 30% of ear fertile part (Table 7). Finally, no differences between the two environments were found per each of the other treatments (DI75, DI50, and DI25) in terms of yield.

Regarding the plant height, under PE conditions, it was always higher than that of the open field plants (Table 6). Plants of the PE-DI100 treatment showed an increase of about 9.5% compared to the corresponding plants grown in the open air, but without significant differences among them. Only the two DI50 treatments were significantly different between them, and a stronger difference between the two environmental conditions was also recorded in the DI25 treatment, with a 18.7% increase under PE with respect to the OF. As for ears' parameters (Table 7), the basal diameter of ears of open field plants was significantly higher than tunnel plants (2.6 vs. 2.4 cm), and the only statistical differences between the two environments were found for the DI100 treatment. The ears' length decreased when water stress increased in both environments, without significant differences between them. Awe et al. [39] observed that the ear length in the water control treatment was higher than in the water deficit treatments (0.75IE, 0.50IE, and 0.35IE), which were statistically not different. In early studies on the application of deficit irrigation on maize, yield reduction was reported [40,41] due to a decreased number of ears per plant [42]. Particularly, Cakir [8] highlighted that water stress application during the vegetative stage mainly reduced ear numbers per square meter.

Water use efficiency (WUE) decreased with increasing water stress; for DI0, it was not possible to calculate the WUE because the plants in this treatment did not produce grain. WUE of all treatments under OF conditions was higher than that under PE, but only DI75 treatments were significantly different between them (Table 8), probably due to the higher grain yield recorded for these plants in the open field (Table 6). In addition, contrary to PE conditions, in open field, DI75 was not different from the well-watered treatment (DI100).

In maize, Howell and co-authors [43] reported that WUE increased when the irrigation amount increased, in line with our results. Additionally, in another study, the WUE value decreased as water stress increased, both in cotton and corn cultivated in Turkey [44].

The WUE can increase, decrease, or remain unchanged under water deficit conditions, depending on the genotype and the water stress level [45]. According to the literature, increasing WUE under water stress conditions indicates an adaptation to deficit irrigation, while a WUE reduction occurred in sensitive hybrids of maize [46].

**Table 6.** Interaction between environment (PE = high temperatures and OF = ordinary temperatures) and water treatment (DI100 = 100%, DI75 = 75%, DI50 = 50%, DI25 = 25%, and DI0 = 0% restoration of water lost by evapotranspiration) on yield, plant height, and number of ears.

| Irrigation | Yield (t ha$^{-1}$) | | Plant Height (cm) | | Ears (No. m$^{-2}$) | |
|---|---|---|---|---|---|---|
| | PE | OF | PE | OF | PE | OF |
| DI100 | 9.9 a | 8.5 b | 146.6 a | 133.8 ab | 13.0 a | 11.0 b |
| DI75 | 5.8 c | 6.7 c | 123.2 bc | 114.4 c | 10.0 bc | 9.0 cd |
| DI50 | 1.7 d | 1.7 d | 90.8 d | 78.8 e | 8.0 de | 7.0 e |
| DI25 | 0.8 de | 0.4 e | 58.4 f | 49.2 f | 1.0 f | 1.0 f |
| DI0 | 0.0 e | 0.0 e | 35.7 g | 36.7 g | 0.0 f | 0.0 f |
| **Significance** | | | | | | |
| Irrigation (DI) | ** | | ** | | ** | |
| Environment (E) | NS | | NS | | NS | |
| DI × E | ** | | ** | | ** | |

NS, **refer to not significant or significant at $p \leq 0.01$, respectively. Different letters within each column indicate significant differences according to Duncan's multiple range test.

**Table 7.** Interaction between environment (PE = high temperatures and OF = ordinary temperatures) and water treatment (DI100 = 100%, DI75 = 75%, DI50 = 50%, DI25 = 25%, and DI0 = 0% restoration of water lost by evapotranspiration) on basal diameter, length, and fertile part (F.P.—% on total length) of ears.

| Irrigation | Basal Diameter (cm) | | Length (cm) | | F.P. (% on Total Length) | |
|---|---|---|---|---|---|---|
| | PE | OF | PE | OF | PE | OF |
| DI100 | 3.2 b | 3.7 a | 17.0 | 16.5 | 76.0 a | 77.4 a |
| DI75 | 3.1 b | 3.5 ab | 15.3 | 15.3 | 73.9 b | 75.7 ab |
| DI50 | 1.9 c | 2.2 c | 10.7 | 10.4 | 54.2 d | 67.5 c |
| DI25 | 1.3 d | 1.2 d | 5.0 | 4.2 | 27.8 e | 26.7 e |
| DI0 | nd | 0.6 e | nd | 2.4 | nd | 0.0 f |
| **Significance** | | | | | | |
| Irrigation (DI) | ** | | ** | | ** | |
| Environment (E) | NS | | NS | | NS | |
| DI × E | * | | NS | | * | |

NS, * and ** refer to not significant or significant at $p \leq 0.05$, and 0.01, respectively; nd not detected. Different letters within each column indicate significant differences according to Duncan's multiple range test.

**Table 8.** Interaction between environment (PE = high temperatures and OF = ordinary temperatures) and water treatment (DI100 = 100%, DI75 = 75%, DI50 = 50%, DI25 = 25%, and DI0 = 0% restoration of water lost by evapotranspiration) on water use efficiency (WUE).

| Irrigation | WUE (kg m$^{-3}$) | |
|---|---|---|
| | PE | OF |
| DI100 | 2.49 a | 2.56 a |
| DI75 | 1.94 b | 2.70 a |
| DI50 | 0.86 c | 1.02 c |
| DI25 | 0.82 c | 0.90 c |
| DI0 | - | - |
| **Significance** | | |
| Irrigation (DI) | ** | |
| Environment (E) | * | |
| DI × E | * | |

NS, * and ** refer to $p \leq 0.05$ and 0.01, respectively. Different letters within each column indicate significant differences according to Duncan's multiple range test.

The interactions between environment and water treatment on total nitrogen, nitrogen use efficiency (NUE), and protein percentage of kernels are reported in Table 9. As far as it concerns the total nitrogen content of the vegetative biomass, the values had an

increasing trend, reaching the maximum with DI25, and then decreasing with DI0, but only significantly in the OF. The NUE decreased when water stress increased, and significant differences between the two environments were recorded only for the DI100 and DI75. Probably, the plants of DI50 and DI25, although taking up a greater amount of nitrogen (higher N total content), mainly used it for sustaining vegetative development and not the yield, and for accumulating nitrogen in grains. Indeed, the protein content increased significantly with increasing water deficiency, and the highest percentage was observed in DI25 under open field conditions, which was significantly different from all the other treatments. Our data confirm previous results on the negative effects of a decreased irrigation water amount on the quality of sweet corn [46,47]. Oktem [48] suggested that the protein content in the seeds decreases according to the amount of starch of the kernels, but only with good water availability; indeed, the protein content in kernels could be increased under the water deficit due to low starch content in the kernel, as well as because of lower yield that probably caused a greater concentration of nitrogen in kernels.

**Table 9.** Interaction between environment (PE = high temperatures and OF = ordinary temperatures) and water treatment (DI100 = 100%, DI75 = 75%, DI50 = 50%, DI25 = 25%, and DI0 = 0% restoration of water lost by evapotranspiration) on total nitrogen, nitrogen use efficiency (NUE), and percentage protein into grain.

| Irrigation | N (%) | | NUE (t kg$^{-1}$) | | Protein (%) | |
|---|---|---|---|---|---|---|
| | PE | OF | PE | OF | PE | OF |
| DI100 | 4.75 f | 5.11 de | 0.062 a | 0.053 b | 10.75 e | 12.00 d |
| DI75 | 4.91 ef | 5.32 d | 0.036 d | 0.042 c | 13.20 c | 13.13 c |
| DI50 | 6.04 c | 6.20 bc | 0.011 e | 0.011 e | 13.13 c | 13.75 b |
| DI25 | 6.71 a | 6.80 a | 0.005 f | 0.003 f | 13.75 b | 14.94 a |
| DI0 | 6.48 ab | 6.34 bc | nd | nd | nd | nd |
| **Significance** | | | | | | |
| Irrigation (DI) | * | | ** | | * | |
| Environment (E) | NS | | NS | | * | |
| DI $\times$ E | ** | | ** | | * | |

NS, * and **, refer to not significant or significant at $p \leq 0.05$, and 0.01, respectively; nd not detected. Different letters within each column indicate significant differences according to Duncan's multiple range test.

### 3.4. Leaf Gas Exchanges and Leaf Water Potential Measurements

Plants may counteract the negative effects of drought stress by altering their responses at morphological, physiological, and biochemical levels [49,50]. Leaf growth is sensitive to water stress, and may be reduced even by a limited decrease of soil water potential [51].

Regardless of the environmental conditions, deficit irrigation levels strongly affected leaf gas exchanges' measured parameters, with a linear reduction of net photosynthetic rate (Pn), stomatal conductance (gs), and leaf transpiration rate (E) as the amount of water reintegration decreased from DI100 to DI25 (Table 9). Non-detectable values were collected by non-irrigated plants (DI0). The dramatic reduction of Pn in DI25 plants (−86%) compared to the control was mainly driven by stomatal regulation (−88% in gs). Stomatal closure is known to be the first response of plants to water deficits, preventing the water losses occurring by transpiration [52]. Thus, it is considered as the first level of defense against tissue dehydration, since it happens rather quickly compared to any other possible change in root growth, leaf area, chloroplast structure, and pigment content. Additionally, the midday leaf water potential ($\psi_{lmd}$) was significantly affected by irrigation treatments, with DI25 values lower than DI100 by 31% (Table 9). On the contrary, the intrinsic water use efficiency (WUEi) reached higher values at DI75 (+7%) and DI50 (+16%), and lower at DI25 (−28.4%), compared to DI100. Under mild drought stress, stomatal closure has a protective role in saving the water loss and increasing water use efficiency, whereas, under severe drought stress, it becomes dangerous [53]. Indeed, at 25% of plant water requirement (DI25), the sub-stomatal $CO_2$ concentration dramatically increased compared to the control (nearly 7-fold higher).

Leaf photosynthetic rate and leaf area are strictly related [54], since under drought stress, cell division and cell elongation are decreased, resulting in a reduction of leaf area (Table 3). It is known that under water stress conditions, maize plants adopt the reduction of leaf area as an adaptive strategy. Considering this fact, leaf area index has always been taken into account in the maize breeding programs for drought tolerance [55]. However, if the maize water requirement is reduced with the leaf area decrease, with the aim to sustain plant survival under limited water availability [56], on the other hand, the limitations imposed to photosynthetic activity affect the grain yield [57,58].

Depending on the particular drought circumstances, usually, plants have several strategies to overcome the water deficit stress [59,60]. Indeed, some water conservation strategies are planned to decrease cumulative transpiration and to allow water accessibility for the plants by controlling stomatal conductance and by reducing the leaf evaporative area, other than slow leaf growth rate and early leaf senescence. Under severe drought conditions, such strategies can have advantageous effects, allowing plants to survive. On the other hand, since they imply a reduction of the net photosynthetic rate, thus decreasing the accumulation of biomass, they may impose a significant yield penalty under mild and moderate water deficits. Furthermore, under mild to moderate water deficits, regimes that support high stomatal conductance and continuance of high photosynthesis can be useful, due to their ability to sustain plant growth during and soon after the stress. Additionally, high stomatal conductance by counteracting an increase in leaf temperature can reduce the heat stress effects generally occurring under drought conditions [61]. Recently, several studies on drought stress effects on wheat productivity highlighted how plant responses to water stress can be modulated under changing environmental conditions, particularly, an increasing air $CO_2$ concentration [4,62].

In the present experiment, there was a main significant effect of environmental conditions of cultivation on the measured physiological parameters for Pn, gs, and ψlmd. In detail, midday leaf water potential was significantly lower under the protected environment (PE), but plants reached Pn and gs values increased by 22.5% and 27% respectively, compared to the plants in the open field (OF) (Table 10), which could also be related to even a slight but significant increase in air $CO_2$ concentration under PE compared to open field (382 vs. 371 ppm, respectively).

Water potential can vary noticeably, depending on the drought stress level and on environmental conditions. However, in our study, the daily course of leaf water potential measured in the morning, at midday, and in the afternoon had a similar pattern when comparing the environmental condition, with only the DI25 treatment reaching the lowest levels under both the tunnel and in open field conditions (Figure 4A,B).

**Table 10.** Main effects of deficit irrigation application treatment (DI100 = 100%, DI75 = 75%, DI50 = 50%, DI25 = 25%, and DI0 = 0% restoration of water lost by evapotranspiration) and environmental conditions (PE = high temperatures and OF = ordinary temperatures) on leaf gas exchange and midday leaf water potential of maize potted plants (13 June 2017).

| Treatments | Pn | Cì | gs | E | WUEi | $\psi_{lmd}$ |
|---|---|---|---|---|---|---|
| | $\mu mol\ m^{-2}\ s^{-1}$ | $\mu mol\ mol^{-1}$ | $mmol\ m^{-2}\ s^{-1}$ | $mol\ m^{-2}\ s^{-1}$ | $\mu molCO_2\ molH_2O^{-1}$ | MPa |
| Irrigation (DI) | | | | | | |
| DI100 | 24.75 a | 25.00 c | 165.07 a | 4.54 a | 151.93 ab | −1.69 a |
| DI75 | 19.85 b | 24.63 c | 126.98 b | 3.75 b | 162.61 ab | −1.72 a |
| DI50 | 11.05 c | 58.84 b | 65.78 c | 2.34 c | 176.49 a | −1.93 ab |
| DI25 | 3.36 d | 170.18 a | 19.73 d | 0.88 d | 126.35 b | −2.22 b |
| DI0 | nd | nd | nd | nd | nd | −2.97 c |
| Environment (E) | | | | | | |
| PE | 16.24 a | 76.39 a | 105.74 a | 2.72 a | 145.99 a | −2.23 a |
| OF | 13.26 b | 62.93 a | 83.04 b | 3.04 a | 162.69 a | −1.99 b |
| Significance | | | | | | |
| DI | *** | *** | *** | *** | * | *** |
| E | * | NS | * | NS | NS | * |
| DI × E | NS | NS | NS | NS | NS | NS |

NS, * and *** refer to not significant or significant at $p \leq 0.05$, and 0.001, respectively; nd not detected. Different letters within each column indicate significant differences according to Duncan's multiple range test.

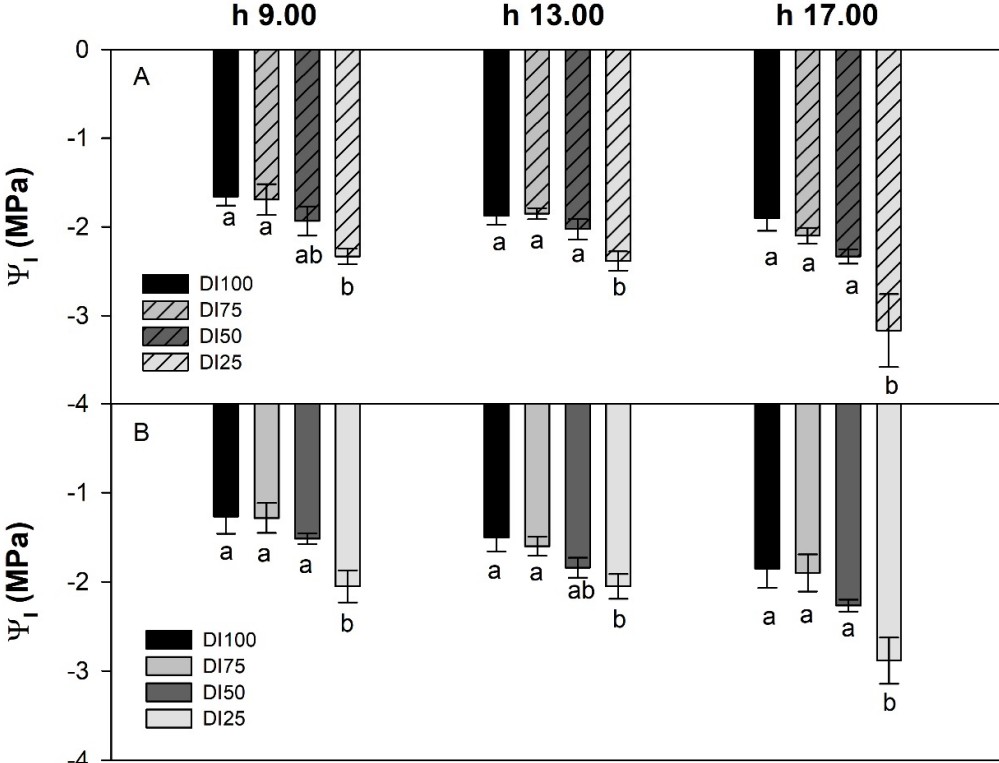

**Figure 4.** Daily course of leaf water potential of maize potted plants grown under four irrigation levels (DI100, DI75, DI50, and DI25) in high-temperature (**A**) or ordinary temperature (**B**) conditions. Different letters within each column indicate significant differences according to Duncan's multiple range test ($p \leq 0.05$).

## 4. Conclusions

Coping with climate change is one of the main challenges of agriculture, driving research toward continuous attempts to identify the best strategies for crop adaptation to these different environmental conditions. Our results highlighted a good adaptability of this early maize hybrid to increased air temperatures (about 3 °C above the current open field condition). Indeed, under tunnel cultivation, the best yield performances were obtained. On the other hand, plants resulted as much more sensitive to drought, as noticed by the growth responses under increased water deficit stress. Furthermore, the crop WUE decreased as the water stress increased, according to the reported behavior of more sensitive plants. However, plants with low WUE are usually more competitive in arid environments because they consume more resources more rapidly, thus suppressing their competitors.

Thus, although this crop in the present experimental conditions appears to actuate some short-time defense mechanisms to counteract moderate and severe water stress, such as the lowering of leaf water potential and the stomatal closure regulation, it resulted in being very sensitive to severe water deficits. On the other hand, this maize hybrid could be suitable in cultivation conditions of Mediterranean areas, where a correct management of amount and time intervals of irrigation can allow to reduce irrigation water requirements despite frequent heatwave events.

Therefore, considering our results, it can be assumed that we could adapt the maize to future climate change.

**Author Contributions:** Conceptualization, M.M., L.O., E.C. and C.C.; methodology, I.D.M.; software, L.O.; validation, M.M., I.D.M. and C.C.; formal analysis, L.O.; investigation, E.C.; resources, M.M.; data curation, E.C.; writing—original draft preparation, L.O., I.D.M. and C.C.; writing—review and editing, I.D.M. and C.C.; visualization, E.C.; supervision, M.M.; project administration, M.M. and C.C.; funding acquisition, M.M. All authors have read and agreed to the published version of the manuscript.

**Funding:** This research received no external funding.

**Institutional Review Board Statement:** Not applicable.

**Informed Consent Statement:** Not applicable.

**Data Availability Statement:** The datasets generated for this study are available on request to the corresponding author.

**Acknowledgments:** We would like to thank Sabrina Nocerino and Roberto Maiello for their support in laboratory work.

**Conflicts of Interest:** The authors declare no conflict of interest.

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
