# Peer review of "Yield Performance and Physiological Response of a Maize Early Hybrid Grown in Tunnel and Open Air under Different Water Regimes"

_sustainability, doi:10.3390/su132011251_

Round 1
Reviewer 1 Report
The adaptability of an early maize hybrid produced under two different temperature settings and with varying water regimes was assessed in this study. The experimental design was a randomized complete block design with two distinct temperature conditions: I normal temperature – in the open field, and ii) elevated temperature (about 3°C above the current condition) – in a poly-ethylene tunnel. The manuscript is well structured and well discussed. However, some points should be checked and corrected before its acceptance in this journal.
- The study's background should be clearly stated. Describe the introduction and review of the work.
- Lack of statistical results. Statistical significance should be provided in figures.
- Please speculate on the results. The discussion must improve.
- The MS English needs to be improved. The article's English must be carefully checked for grammatical errors.
Author Response
Reviewer 1
The adaptability of an early maize hybrid produced under two different temperature settings and with varying water regimes was assessed in this study. The experimental design was a randomized complete block design with two distinct temperature conditions: I normal temperature – in the open field, and ii) elevated temperature (about 3°C above the current condition) – in a poly-ethylene tunnel. The manuscript is well structured and well discussed. However, some points should be checked and corrected before its acceptance in this journal.
Dear Reviewer,
Thanks very much for your reviewing work and for your suggestions, which will improve our paper. You can find the revisions in track changes in the text, the added references in blue color, and below the responses point by point to your comments in bold.
The study's background should be clearly stated. Describe the introduction and review of the work.
We slightly modified the introduction, adding new references, which better focus on the treated issue.
Lack of statistical results. Statistical significance should be provided in figures.
Figure 4, which didn’t report the statistical significance, has been replaced with two new tables (4 and 5), in which the statistical differences are reported.
Please speculate on the results. The discussion must improve.
Thanks for the suggestion, we deeply modified the results and discussion.
The MS English needs to be improved. The article's English must be carefully checked for grammatical errors.
The paper has been revised for the English language
Reviewer 2 Report
The manuscript entitled; Yield Performance and Physiological Response of a Maize Early Hybrid Grown in Tunnel and Open Air Under Different Water Regimes by Ottaiano et al. describes the interesting story about performance of maize under water and heat stress conditions by linking yield traits with water use efficiency and eco-physiology. Although, manuscript is well written, yet author did not describe the results in scientific manner which is the main concern from my side. I will suggest the following changes before its acceptance.
Abstract
The results are not clearly mentioned in the abstract
Introduction
Line 39, please use the following latest reference
Shokat S, Novák O, Široká J, Singh S, Gill KS, Roitsch T, Großkinsky DK, Liu F. (2021). Elevated CO2 modulates the effect of heat stress responses in Triticum aestivum by differential expression of isoflavone reductase-like (IRL) gene. Journal Experimental Botany. DOI 10.1093/jxb/erab247 (In press).
Please state the concrete hypothesis to achieve the objective.
Materials and methods
Line 96, please remove the space between “to measure”
Line 113, please delete the word “each”
Results and Discussion
Although significance levels are indicated in the tables, but authors did not mention significance in the results section. It should be indicated in the text as well. Headings of tables and figures should be self-explanatory. Further, authors can take help from
Ulfat A, Shokat S, Li X, Fang L, Großkinsky DK, Majid SA, Roitsch T, Liu F. (2021). Elevated carbon dioxide alleviates the negative impact of drought on wheat by modulating plant metabolism and physiology. Agricultural Water Management. 10.1016/j.agwat.2021.106804 (In press).
Or
Shokat S, Großkinsky DK, Liu F. (2021. Impact of elevated CO2 on two contrasting wheat genotypes exposed to intermediate drought stress at anthesis. Journal of Agronomy and Crop Science. 207: 20-33.
Conclusion
Line 369, please remove the word “current”
Author Response
Reviewer 2
The manuscript entitled; Yield Performance and Physiological Response of a Maize Early Hybrid Grown in Tunnel and Open Air Under Different Water Regimes by Ottaiano et al. describes the interesting story about performance of maize under water and heat stress conditions by linking yield traits with water use efficiency and eco-physiology. Although, manuscript is well written, yet author did not describe the results in scientific manner which is the main concern from my side. I will suggest the following changes before its acceptance.
Dear Reviewer,
Thanks very much for your reviewing work and for your suggestions, which will improve our paper. You can find the revisions in track changes in the text, the added references in blue color, and below the responses point by point to your comments in bold.
Abstract
The results are not clearly mentioned in the abstract.
This part of the abstract has been modified.
Introduction
Line 39, please use the following latest reference
Shokat S, Novák O, Široká J, Singh S, Gill KS, Roitsch T, Großkinsky DK, Liu F. (2021). Elevated CO2 modulates the effect of heat stress responses in Triticum aestivum by differential expression of isoflavone reductase-like (IRL) gene. Journal Experimental Botany. DOI 10.1093/jxb/erab247 (In press)
Thanks for the suggestion, we added this reference.
Please state the concrete hypothesis to achieve the objective.
Done
Materials and methods
Line 96, please remove the space between “to measure”
Done
Line 113, please delete the word “each”
Done
Results and Discussion
Although significance levels are indicated in the tables, but authors did not mention significance in the results section. It should be indicated in the text as well. Headings of tables and figures should be self-explanatory. Further, authors can take help from
Ulfat A, Shokat S, Li X, Fang L, Großkinsky DK, Majid SA, Roitsch T, Liu F. (2021). Elevated carbon dioxide alleviates the negative impact of drought on wheat by modulating plant metabolism and physiology. Agricultural Water Management. 10.1016/j.agwat.2021.106804 (In press).
Or
Shokat S, Großkinsky DK, Liu F. (2021. Impact of elevated CO2 on two contrasting wheat genotypes exposed to intermediate drought stress at anthesis. Journal of Agronomy and Crop Science. 207: 20-33.
Thanks, we deeply modified the results and discussion
Conclusion
Line 369, please remove the word “current”
Done